# Locust and Grasshopper Outbreaks in the Near East: Review under Global Warming Context

Battal Çiplak 

Department of Biology, Faculty of Science, Akdeniz University, 07058 Antalya, Turkey; ciplak@akdeniz.edu.tr

**Abstract:** Plagues of locust possibly date back to before humanity, as they evolved before humans. Following the Neolithic revolution and the permanent settlement of humans in Mesopotamia, locusts and grasshoppers have become a serious problem for people, as imprinted on archaeological remains. In the Near East, desert locust may be an important problem during invasion periods, in addition to various local species of locusts and grasshoppers. Past plagues caused serious disasters in the region, but there has been a pause since the 1960s, thanks to more effective monitoring and control. However, global warming and other anthropogenic activities change ecosystems, and these increase the potential for locust outbreaks, upsurges and plagues for the region. Outbreaks of some local species could also be a serious problem. Pest species of the locust and grasshopper of the Near East mainly belong to Caelifera and some to Ensifera. Global warming and extended agricultural activities can increase the potential for outbreaks of local species and create suitable conditions for desert locust invasions. This review is an attempt to (i) provide a historical background for locust invasions/outbreaks in the Near East, (ii) assess the potential for outbreaking of local species and (iii) define a perspective for future actions regarding global changes.

**Keywords:** locust swarms; Near East; Mesopotamia; swarming potential of resident species; *Schistocerca greraria*; *Dociostaurus maroccanus*

## 1. Introduction

Phylogenetically, the origin of the desert locust, *Schistocerca gregaria* Forskål, dates back to around eight million years ago [1], but that of humans as the genus *Homo* to date back to 2.5 million years and modern human *Homo sapiens* to about 200 thousand years [2]. As desert locust and *Homo sapiens* are both native to Africa, we can speculate that modern humans evolved under the selection pressures of the disasters caused by the desert locust plagues. Whether or not *Homo sapiens* has managed to adapt to this selection pressure is a provocative question, but the available data points to the side of "no". For instance, in recorded history, there are indications of a locust problem in Assyrians and New Testament of the Bible [3–5]. All these histories come from the Middle East and this study aims to present a review of the locust problem in a part of this region, starting from Sinai in the south, extending as far as to the Caucasus in the north, to the Aegean Sea in the west and Zagros Mountain range in the east (Figure 1). The southern part of the Arabian Peninsula is left out because this region is within the recession area of the desert locust and has already received considerable interests from locust experts (for a review see [6]). Comparing to North Africa and Southern part of the Arabian Peninsula, the locust and grasshopper problem in the Near East has other peculiarities, and thus requires a separate evaluation. In this region, the desert locust plagues have been relatively rare. The area is outside recession range of the species and is only contaminated during significant plague periods [7]. In addition, there have been no desert locust swarms in the region since 1960s. However, there are other outbreaking species, such as *Dociostaurus maroccanus* (Thunberg), *Locusta migratoria* Linnaeus and some species of *Calliptamus* Serville. Moreover, there have been occasionally and locally outbreaking species, such as *Heteracris pterosthica* (Fischer

de Waldheim), *Notostaurus anatolicus* (Krauss) and *Arcyptera labiata* (Brulle), especially in Anatolia, and an assessment of their potential in the context of global warming seems of particular importance. Some species of Barbististini—*Isophya* Brunner von Wattenwyl, *Poecilimon* Fischer and *Phonochorion* Uvarov—will also be considered, although none of these long-horned Orthoptera could be considered a locust and there is no significant report on their outbreaks. The review will begin by summarizing the locust problem in the previous century, especially before the 1960s. In the second section, common and locally outbreaking species as well as their management will be considered, especially in Anatolia and Mesopotamia. Finally, an assessment of the possible impact of global warming and other anthropogenic activities on the locust and grasshopper problem will be discussed. The conclusion aims to provide a management perspective.

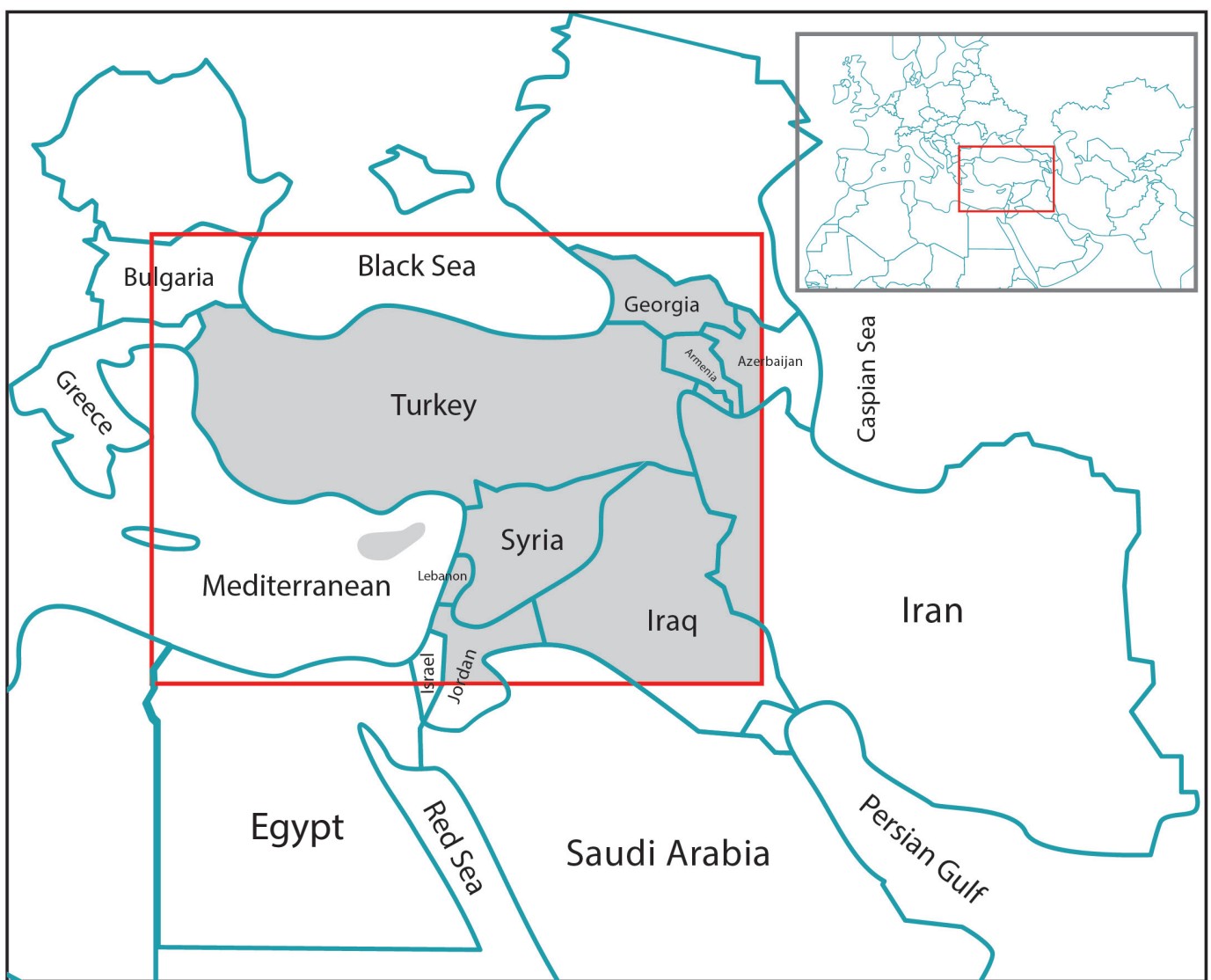

**Figure 1.** The geographic area considered in this review.

## 2. History of the Locust and Grasshopper Problem in the Near East—First Half of the 20th Century

Considering the Near East (Figure 1), publications on locust plagues in the 20th century can easily be assembled into two groups according to their dates. Publications included in the first group are rarely after 1960 [8–18] and if so, they review plague events of the past [19]. Rainey et al. [20] clearly defines ending of a period as no plagues reached to this

region after 1962. A new period of the studies begins after 2000, although the publications by orthopterists or locust experts are scant among them (e.g., 6). The new period is mainly characterized by publications of historians [21–32], also dealing with locust invasions of the first half of the 20th century. They combine information from archives of governments and old newspapers and provide a significant amount of knowledge about the history of locust invasions in the Near East. For example, thanks to these publications, we discovered the acceptance of a special law for locust control by the Ottoman Parliament in 1912, which seems to be a unique event in this regard [28,29]. The law consists of nine articles. The first article defines responsibilities of local peoples in monitoring their district and informing the local governmental officers and regulate awarding of peoples who supply accurate information. The second article defines how to treat the areas contaminated with eggs and hoppers. The third article lists responsibilities of persons in a hierarchical way in destroying the eggs or hoppers. Later, six articles define the rules for offices and staffs of the government, the budget to be provided and its usage. This law was updated in subsequent years by Parliaments of Ottoman and Turkish Republic. Almost all of these publications report or discuss the plagues caused by two species—*S. gregaria* and *D. maroccanus*—indicating they were main locust pests in this area, and these will mainly be considered below. Damage to agriculture by other species such as *Calliptamus italicus* (Linnaeus) (or other species of the genus), *Gryllus campestris* Linnaeus, *Platycleis intermedia* (Serville), *Platycleis affinis* Fieber and *H. pterosthica* are negligible [14,16,29].

### 3. Desert Locust Plagues in the Near East

The range of the desert locust is variable depending on plague period. Regarding the present recession area [7], the desert locust range is between latitudes 10° N–35° N and longitudes 15° W–72° E, mainly associated with the Sahara Desert in North Africa, the Arabian Desert in the Arabian Peninsula and the Thar Desert in Pakistan/India. However, the extent of the range can double during periods of invasion. The Near East remains outside the recession area and constitutes a marginal part of the of invasion area [6,7,33]. The history of desert locust plagues during the last two centuries was summarized in Table 1 and the maximum northward extension of the invasions during the 20th century was shown in Figure 2. The extent of the past plagues in the Near East depended on some other peculiarities such as the severity of the plague, rain regime and the control efforts. Several desert locust plagues were reported for the period before the First World War (Table 1) [19,23,33], in particular for Mesopotamia and the Levant. Although data on the severity and extent of plagues are limited and mainly concern human aspects due to the destruction of food resources and control management by the Ottoman Government [12,22,31], they allows us to estimate that at least the Levantine and the Mesopotamian regions have been invaded on several occasions. According to reports, possibly the greatest plague occurred during First World War and its peak was in 1915 and ended in 1917. As controlling actions were primitive and ineffective, such as destroying the locusts by hand (Figure 3); its consequences were catastrophic. It resulted in a great famine, which was considered one of the greatest tragedies in the region, with the death toll attributed to it estimated at around 500,000 in 1918 [15,24,29–31]. Existing records, based on certain governmental documents, indicate that there was also a significant control effort (see [12,23,29,31] and references therein), but not sufficient to prevent a disaster. During these years, a local reproduction regularly took place, particularly in the Levant and Mesopotamia, leading to successive outbreaks and increasing the impact of the plague.

**Table 1.** History of *Schistocerca gregaria* plagues in the last two centuries (those in the 20th century are shown in Figure 2).

| Year/Period | Invasion Area in Near East | Reference |
|---|---|---|
| 1729 | Aleppo | [22,32] |
| 1865 | Levant (from Sinai to Turkey), including Cyprus | [19] |
| 1878 | Sinai, Palestine, Syria (Levant) | [19] |
| 1890 | Sinai, Palestine, Syria (Levant) | [19] |
| 1902 | Sinai, Palestine, Syria (Levant) | [19] |
| 1911–1915 | Arabian Peninsula plus Mesopotamia (Syria, Lebanon, Iraq and Turkey) up to Southern Caucasian Plateau | [8,12,15,19,24,29,31] |
| 1928–1930 | Arabian Peninsula plus Mesopotamia (Syria, Lebanon, Iraq and Turkey) up to Southern Caucasian Plateau | [8,19,29] |
| 1945 | Arabian Peninsula plus Mesopotamia (Syria, Lebanon, Iraq and Turkey) up to Southern Caucasian Plateau | [8,19] |
| 1952 | Arabian Peninsula and a very small part of the Mesopotamian Turkey | [8,19] |
| 1953 | Arabian Peninsula and large parts of the East plus South East Turkey | [8,19] |
| 1958 | Arabian Peninsula and large parts of the East plus South Turkey | [19] |
| 1959 | Arabian Peninsula plus a small part of Mesopotamian Turkey | [19] |
| 1960 | Arabian Peninsula plus a small part of Mesopotamian Turkey | [19] |
| 1962 | Arabian Peninsula plus Mesopotamian Turkey | [19] |

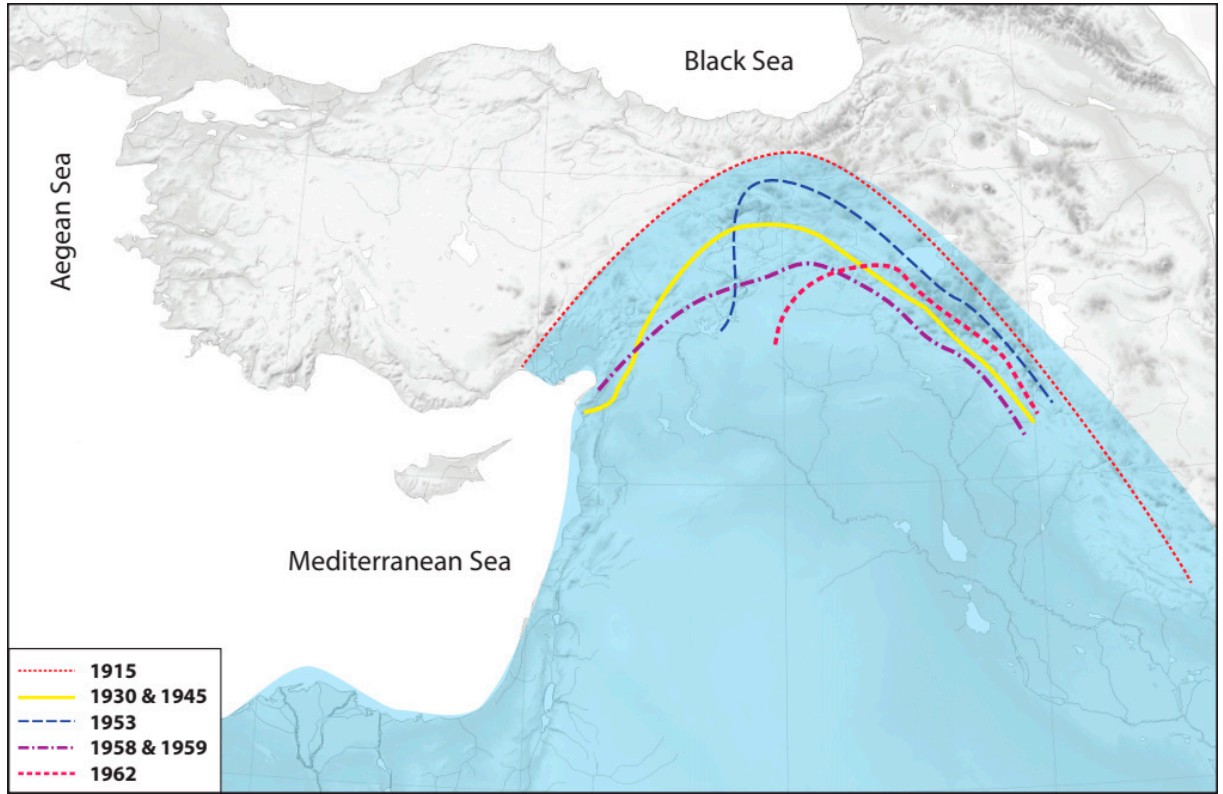

**Figure 2.** Northern invasion borders of *Schistocerca gregaria* plagues in the 20th century in the Near East (figure prepared using data by Balamir [19]).

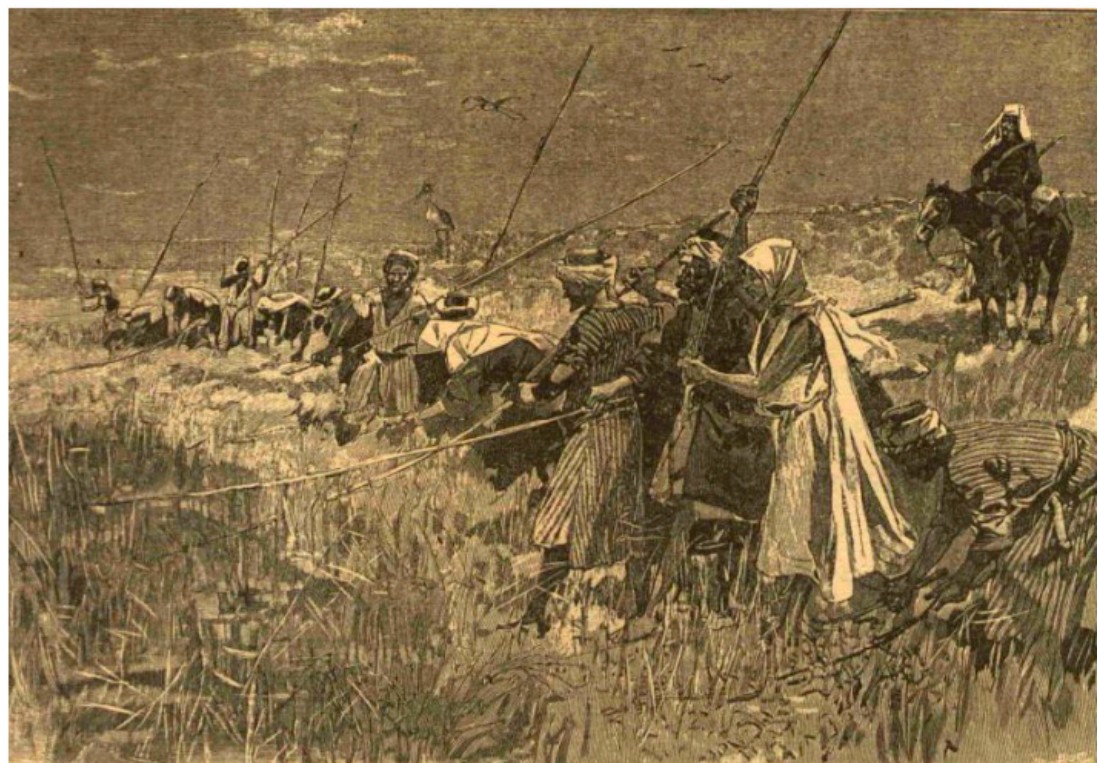

**Figure 3.** Locust destruction by sticks at hand in Palestine (from the newspaper *Serveti Fünun*, 269, 1896, page 140).

Balamir [19] presented an extensive review of desert locust invasions in the Near East after the First World War, including extent, control activities and consequences of the invasions. Eight different plagues have been reported in the years 1930, 1945, 1952, 1953, 1958, 1959, 1960 and 1962 (Table 1). The size of these invasions and their northern boundaries were variable (Figure 2). The plagues extended to northern Syria and Iraq, but not to Turkey (or only into a small part), in 1952, 1959 and 1960. In 1962, the plagues invaded the Mesopotamian lowlands of Turkey, as well as, further south, the Levant, Jordan and Iraq. In the remaining years, plagues reached to mountain belts in the northern Mesopotamia. The two most extensive plagues were those of 1930 and 1958, and the invasion margin reached the Caucasus Plateau in north-eastern Turkey. Various preventive actions were taken. The use of sheet metal dams (made of zinc plates) for the management of hopper bands was the main control method. Hoppers were channeled into ditches and then crushed, burned or buried. For example, a total of 67,690,000 kg and 603,142,000 kg nymphs were destroyed in Turkey by this method during the 1930 and 1958 plagues, respectively [19]. Spraying insecticide (mainly the gamma-hexachlorocyclohexane preparations) by air became a dominant method after mid of the 20th century. Great success was achieved by this method and none of the 20 swarms of 1960 succeeded in laying eggs in Turkey [19]. Other control methods implemented in the past were ploughing egg-laying sites, collecting nymphs and adults by hand or spraying liquid soap or some other chemicals, such as arsenic [19,29,32,33].

There are no records of desert locust swarms in the Near East after the years mentioned above, which corresponds to the general trend in other parts of the invasion area [20]. This is possibly the reason why the countries in the region have halted governmental units for locust control or not established new ones [6]. Why did such a period of successive plagues, observed since at least the last quarter of the 19th century, end in the last half of the 20th century? In fact, after 1965, effective control activities, mainly in Africa and southern Arabia—efficient survey and timely response in desert locust breeding areas—appear to be the main reasons for this improvement and the absence of swarms in the Near East [7,20,34–39]. This is also applicable for the peripheral invasion areas, as using spraying aircrafts after 1950 prevented the spread of the swarms to the borders of Caucasus.

Although, in recent years, some swarms entered in the Near East, none of them were able to become harmful.

### 4. Moroccan Locust Plagues in the Near East

Considering the Near East, one of the most harmful locust species was the Moroccan locusts. This species seems to have recently lost its economic importance to the point that conservation measures have been suggested only for certain localities [40]. However, it was a serious problem for the Near East in the first half of the 20th century, especially important in West Anatolia and moderate highlands of Mesopotamia, including south-eastern Turkey and the northern regions of Iraq and Syria for around a hundred-year period. Several very harmful plagues of this species have been reported in the Near East during the period of 1847–1945 [12,18,25,29,31] (Table 2). Although there are no reliable reports for more archaic time, this locust species may have been a serious problem at all times in the Near East. West Anatolia, Mesopotamia and Southern Caucasus also were three important outbreak areas for this species (Figure 4). The outbreak periods were mostly different, although sometimes overlapping, from one region to another.

Seven plague periods have been reported in West Anatolia for a century, from the mid-19th century to the mid-20th century. Each plague lasted for 2 to 7 years. The plague extent varied from a few km$^2$ to the whole West Anatolia, from the Marmara Sea to the Mediterranean Sea, excluding narrow coastal areas. Some plagues also spread to the European part of Turkey. The three plagues of the second half of the 19th century—1847–1851, 1852–1864 (especially in 1861) and 1875–1881—invaded large parts of West Anatolia [25,29]. The plague in 1904–1905 was restricted to the eastern parts of the Manisa and Uşak provinces of Turkey and possibly played a significant role in the development of the next one, which lasted for about seven years from 1910 to 1917. It was possibly the largest in terms of extent and impact. The upsurge period started in 1909–1910 in Eşme and Kula, then reached its peak in 1916 and declined in 1917. All these stages were well monitored and documented [12,19]. As the Moroccan locust was a persistent problem for West Anatolia, Uvarov [18] was invited to study this species in the region; he considered the 1910–1917 plague as a reference with which to define the bio-ecology of the Moroccan locust. In addition, a considerable effort had been devoted to the prevention of future plagues [12]. Karabağ [16] reported that 125,000,000 kg nymphs and 12,500,000 kg egg pods were destroyed during the period of 1915–1917. Although it was largely controlled in West Anatolia, local swarms in the Marmara region, including parts of Thrace in 1919, were likely the aftershock of this plague period. For the aforementioned reasons, studies on this plague constitute a reference for understanding the upsurges characteristics of species in relation to ecology and for developing effective control strategies. These experiences have provided significant knowledge for the prevention of subsequent plagues. The later plagues, in 1930–1932 and 1939–1941, were also widespread and invaded large parts of West Anatolia up to Marmara Sea in the north, the Konya province-central Anatolia in the east and the north of Antalya in the south. However, both were controlled within one or two years and their damage was limited. Later, especially after 1962, only a few local swarms occurred and these were successfully controlled.

Mesopotamia (Syria, Iraq and Turkey) was another outbreak area for the Moroccan locust in the Near East (Figure 4). There are no data regarding plagues of this species for earlier periods; however, four plague periods are well known in first half of 20th century [12,13,29]. Those in 1919 and 1945 were one-year plagues affecting a restricted part of Mesopotamia. The two others, in 1931–1932 and 1939–1941, lasted at least two years each and invaded larger areas in the region. In addition, the plague of 1925 in Iraq was reported to cause serious damage [40]. The South Caucasus lowlands in the Basins Aras and Kura rivers appear to be a separate outbreak area, and the plague of 1920–1921 in Armenia and Azerbaijan caused significant damage to the region [29,40].

**Table 2.** History of *Dociostaurus moroccanus* plagues in the last two centuries (those in the first half the 20th century are shown in Figure 4).

| Year/Period | Invasion Area | Reference |
|---|---|---|
| 1830 | West Anatolia, local swarms | [25,29] |
| 1847–1851 | Aegean Region and south Marmara | [25] |
| 1852–1864 (peak in 1861) | Large parts of West Anatolia | [25,29] |
| 1875–1881 | Antalya, Inner part of Aegean Region up to Balıkesir Province | [29,31] |
| 1904–1905 | Manisa (inner parts) and Uşak provinces | [25,29,31] |
| 1909 | Local swarm in Eşme in Manisa Province | [18,29,31] |
| 1910–1911 | Swarms in Manisa province (Eşme, Kula, Demirci, Alaşehir) | [13,18,29] |
| 1911–1912 | West Anatolia (Denizli, Manisa, Aydın and İzmir provinces) | [13,18,29] |
| 1915–1916, declined in 1917 | Large parts of western Anatolia up to Antalya and Konya provinces | [13,18,29] |
| 1919 | Marmara Regions of Turkey | [13,18,29] |
| 1919 | A restricted part of Mesopotamian Turkey, Iraq and Syria | [29] |
| 1920–1921 | Armenia and Azerbaijan | [40] |
| 1930–1932 | Large parts of West Anatolia including Marmara Basin plus Konya province in Central Anatolia | |
| 1931–32 | Mesopotamian Turkey (from Siirt to Mersin), Iraq and Syria | [13,29] |
| 1934–1938 | Local swarms in several location especially in West Anatolia | [13,29] |
| 1939–1941 | Aegean and Southern Marmara regions, and local swarms in Mesopotamian Turkey, Iraq and Syria | [13,29] |
| 1945 | A restricted part of Mesopotamian Turkey and Syria | [29] |

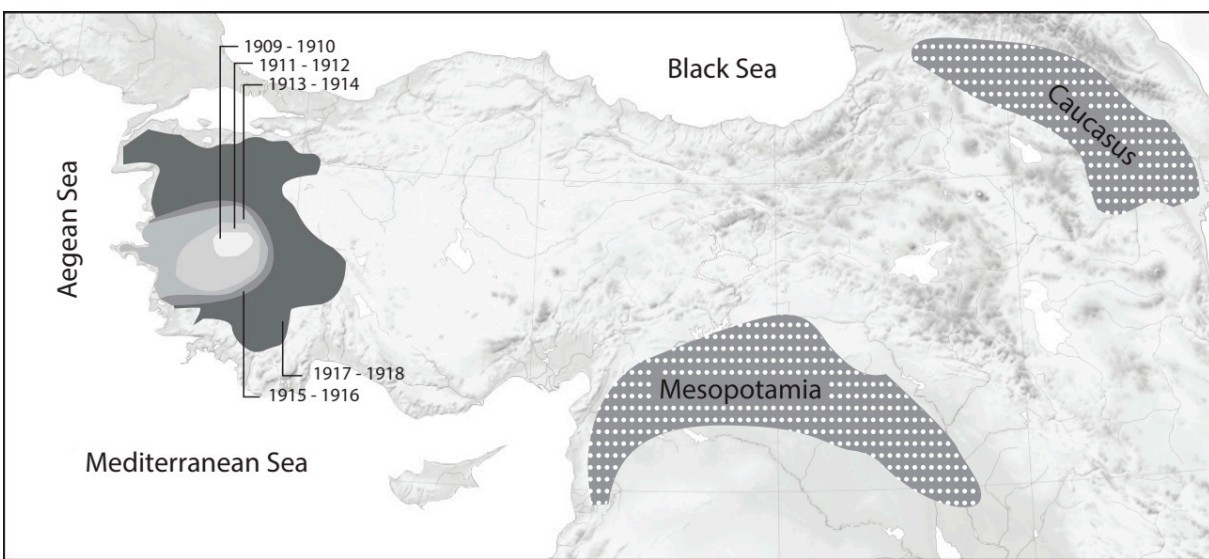

**Figure 4.** Outbreak areas of *Dociostaurus moroccanus* in the Near East: West Anatolia, Mesopotamia and Caucasus. Development of the most intense plague during 1909–1917 in West Anatolia is outlined by years. The borders for 1917–1918 also define the West Anatolian outbreak area. Figure is based on data mainly provided by Bücher [12] and Uvarov [18]).

The Moroccan locust plagues in the Near East exhibit some peculiarities. It seems that they are several independent outbreak areas and these can be classified as West Anatolia, Mesopotamia and the Caucasus (see Figure 4) [12,18,40]. In West Anatolia, the plains of the Inner Aegean Basin, especially those in Manisa, Uşak and the surrounding parts of the neighboring provinces of Denizli and Aydın, were the main outbreak areas.

Swarms that originated in these regions spread in subsequent years and extended to the surrounding parts of Aegean, Marmara, the Mediterranean and central Anatolia (Figure 4). The Mesopotamian outbreak area may include two places of origin: one in northern Syria and the other in northern Iraq. Uvarov [18] stated that the outbreak area of this species in West Anatolia exhibits special environmental conditions: (i) it is restricted to altitudes of 400–1000 m, (ii) it has a certain precipitation regime and (iii) it harbors a well-defined vegetation (the presence of *Plantago* spp. and *Hordeum* spp. is of particular importance). The determinative aspects of these environmental conditions have been reported from other parts of the species range [40]. This combination of factors restricts the species to parts of West Anatolia, but arising upsurges can invade the surrounding parts of the regions in Anatolia.

The Moroccan locust does not occur in Euro-Siberian vegetation along the Black Sea Basin up to the Great Caucasus. The average elevation of central Anatolia is around 1000 m and that of eastern Anatolia is around 1500 m [41], and these heights possibly prevent the species from colonizing these areas. Consistent with this viewpoint, no swarm or outbreak areas have been reported from central and eastern Anatolia, although the solitary form has been recorded frequently [9,42,43]. For similar reasons, the south-eastern Taurus Mountains and the Syrian Desert constitute the northern and southern borders of the Moroccan locust outbreak area in Mesopotamia, respectively. Thus, this outbreak area of the Moroccan locust in this region constitutes an arc starting from Mediterranean Syria and extending to Zagros. The differences between the West Anatolian and Mesopotamian outbreak areas are mainly altitudinal, with altitudes of 400–1000 m and 200–400 m, respectively [14,18], indicating that the latter outbreak area is less suitable. This is possibly the reason why the Moroccan locust plagues in this region were less severe than those in West Anatolia.

The moderate highlands of the inter-montane basins of the Aras and Kura rivers in southern Caucasus were the third important outbreak area in the Near East (Figure 4). Past reports indicate that this is a less significant outbreak area compared to the two others. It should be noted that these three outbreak areas are independent: no swarms in one area spread to another. The periodic overlap of some past plagues of these three distinct areas needs an explanation, but periodic climate changes lasting several years may be the main reason.

An issue worth highlighting is that there has been a pause in the Moroccan locust plagues since the mid-20th century. Although there have been a few local swarms ([44,45]; N. Babaroğlu—personal communication), none of them were invasive and they were controlled in their local outbreak area. This pause may be the result of a combination of several factors. The bio-ecology of the species is surely one of them, as indicated by Uvarov [18] and Latchininsky [40]. The reasons for the pause may be different for West Anatolia, Mesopotamia and the Caucasus. However, the topographic and ecological heterogeneity of the biogeographic Anatolia (see [41]) offers opportunities for the species to compensate for environmental changes. The most plausible reason for the pause seems to be the intense control of swarms in the outbreak areas, preventing upsurges in subsequent years. The swarm in 1996 in Eynif Plain in Akseki (Antalya, Turkey) was controlled by the application of 40,000 kg of insecticide [44,45]. Several similar local swarms occurred in the Aegean part of Turkey (A. Yüzbaş and N. Babaroğlu—personal communication), but none of them succeeded in invading other areas in the following years. This also indicates that control in the outbreak areas is the key reason for preventing long-lasting plagues.

## 5. Other Pest Orthopteran Species

The desert locust and the Moroccan locusts were the two most important agricultural pests in the Near East in 20th century. However, limiting to these two species seems insufficient for the region, especially for Anatolia. Although their characteristics are different from these two species, many other orthopteran pests have been reported in past studies [10,11,14,16,46–48]. We can cite some Acrididae species, such as *L. migratoria*, *Calliptamus* spp. (namely *C. italicus*, *C. barbarus* (Costa) and *C. tenuicercis* (Tarbinsky)), *H. pterosticha*

and *Arcyptera labiata*; Ensifera species, such as *Melanogryllus desertus* (Pallas) and *P. inter-media*; and some species of *Poecilimon*, *Isophya* and *Phonochorion* (without certain specific identification). Of all these species, *L. migratoria* and *C. italicus* were the most important, corresponding to Uvarov's definition of locust [17] and requiring control measures [14,48].

The author himself observed many local swarms of orthopteran species during his field studies since 1990 across Anatolia. As mentioned above, the swarm of *D. maroccanus* in 1996 in the Eynif Plain (near Antalya) was observed from the merging of dense patches of hoppers into larger bands and then into migrating adults. Swarms of other species were also observed, but their swarming characteristics were different from typical locusts. These species present a great taxonomical variety, but mainly belong to Acrididae and Phaneropterinae. Among Acrididae, *L. migratoria*, *C. italicus*, *C. barbarus*, *C. tenuicercis* and *H. pterosticha* are the species reported in previous publications. *N. anatolicus* (Krauss), which coexists mainly with *Calliptamus* spp., is another species to be added to this list. *L. migratoria* is the model species on which Uvarov [17] based his phase theory. Local swarms of the species have been reported ([10]; N. Babaroğlu—personal communication) and control measures have been applied locally by the Turkish Ministry of Agriculture (by central and local plant protection institutions). *L. migratoria* is common in its solitary form and can pullulate in paddy fields, but there is no record of invasive plagues extended to other places. *H. pterosticha* occurs in humid plains irrigated by humans below 1000 m elevation. For example, its population showed a significant increase in 1961–1962 in Diyarbakır and was prevented by the application of chemical insecticide [11]. The author observed a similar case in the province of Malatya in Mesopotamian Turkey. The population density gradually increased in 2017–2020 in the plain part of the province. The increase in population density caused significant damage on crops and fruit trees. Apricot cultivation is the main agricultural activity in the region and these grasshoppers climb to apricot trees and consume both leaves and fruits, as complained about by farmers. This case should be monitored in the coming years. *Calliptamus* spp. and *N. anatolicus* can reach high densities, especially in arid agricultural fields in Central Anatolia, Mesopotamia and some plains of eastern Anatolia. The damage caused by *Calliptamus* spp., especially *C. italicus,* and insecticide application to manage their populations is known for a long time [48]. Although Tutkun ([48]; see also [29,31]) states that the species is the third most harmful locust species in the area, there has been no recent report of outbreaks or preventive control measures. However, monitoring seems to be a necessity.

The case of the phaneropterid species is more interesting. Highly dense populations of *Poecilimon* spp. and *Isophya* spp. have been reported in previous publications [47] or observed by the author himself. Only the two most striking cases are mentioned here. *Isophya hakkarica* Karabağ, whose range covers the south-eastern Taurus and the Anatolian Diagonal [49], reached to very high population densities in the Malatya, Elazığ and Adıyaman provinces of Turkey in 2016. Rural people asked the local government to control the pest as it caused extensive damage to pasture and destroyed animal feed. Residents observed that the increase in population density began in 2015 and peaked in 2016. Local governmental agencies provided partridges to be release into the wild to try to control the pest, but no insecticide was applied. A similar case has been observed with *Poecilimon celebi* Karabag in the Kastamonu province. During a field study in 2012, an extremely dense population was observed in the forest clearing areas with Euro-Siberian steppe vegetation (Figure 5). Locals complained about the destruction of fields and meadows where animal feed was produced. Many other similar cases with species belonging to *Poecilimon*, *Ispohya* and *Phonochorion* (all from Barbitistini) have been observed, especially in northern Anatolia along the Black Sea Basin and the highlands of the eastern Anatolia with a high grassland vegetation. The conclusion is that several Barbitistini species may reach high population densities, cause significant damage and require various control measures.

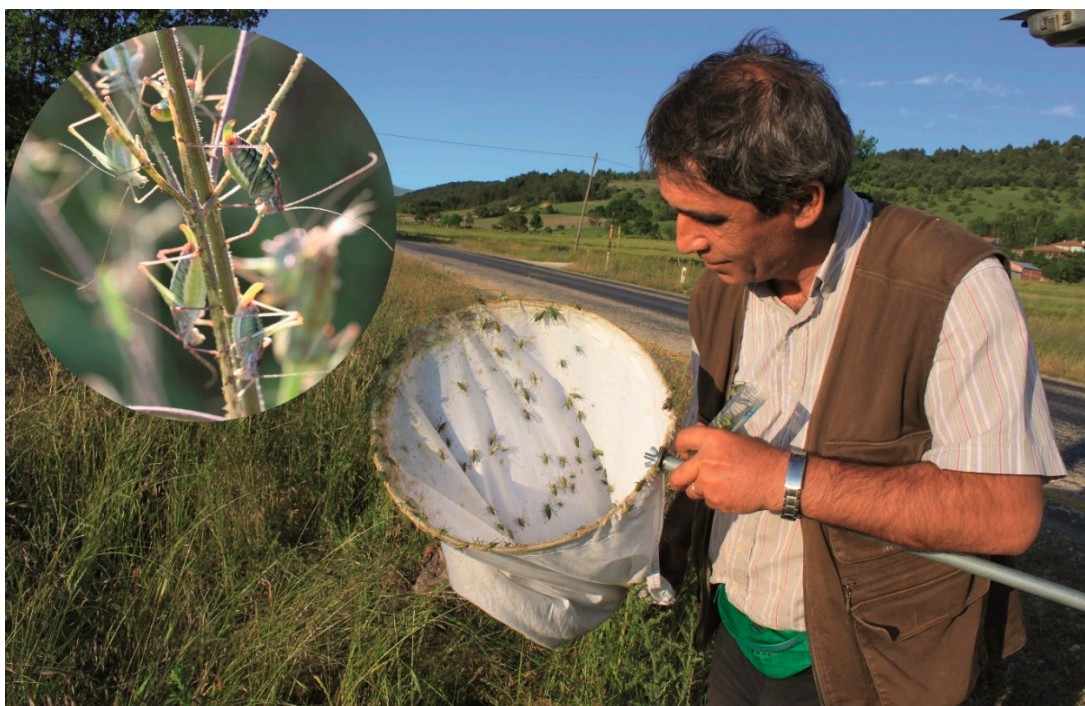

**Figure 5.** Collecting specimens of *Poecilimon celebi* in an outbreak area in North Anatolia.

The reports on the locust and grasshopper management from the Agriculture Ministry of Turkey since the second half of the 20th century, as well as the author's observations during last 30 years, allow us to make some general statements that could guide future plans and actions. Among the previously mentioned species, both *S. gregaria* and *D. moroccanus* show locust characteristics with phase polyphenism (shift from solitary to gregarious phase according to population density), as defined by Uvarov [18,50], and can invade new areas by migrating from the outbreak areas where the initial increase in density occus. *L. migratoria* and *Calliptamus* spp. (specifically, *C. italicus*, *C. barbarus* and *C. tenuicercis*) were also reported as locusts; however, in previous Near East studies or reports, no specific phase changes or long-distance migration leading to invasions of new areas were observed. Instead, they were reported to be harmful only by reaching high densities in their regular distribution range (N. Babaroğlu—personal communications). This statement is undoubtedly valid for the other acridid species: *A. labiata*, *H. pterosticha* and *N. anatolicus*.

The case of long-horned species requires a distinct assessment. All the long-horned or ensiferan species mentioned above have a limited range or are endemic to a specific location. In addition, they are short-winged and lack the ability to migrate. They can exhibit various pigmentation forms, such as green or black as in some *Isophya* and *Phonochorion*, or green and yellow/brown as in some species of *Poecilimon* [51], but these colors do not indicate phase polyphenism due to population density. Although they may pullulate and cause significant damage locally, their outbreaks are only local and population density increases within their regular range. This does not mean that they do not require monitoring and control measures. Several species of Barbitistini and, in addition, some other tettigoniids species can reach unexpectedly high population densities in their local range depending on annual environmental conditions, which may possibly require survey and control measures if a density threshold is exceeded.

## 6. Locust and Grasshopper Management in Turkey in Recent Years

Locust and grasshopper management activities in Turkey are organized under the authority of the Directorate of Plant Protection Central Research Institute (DPPCRI) a branch of the Ministry of Agriculture and Forestry (N. Babaroğlu, M. Çulcu and E. Akci from DPPCRI—personal communication). The main management activity is applying

insecticide over outbreak areas. Releasing partridges is an uncommon practice; there is no report on the efficiency of this method. For the period of 2013–2020, the control of grasshopper outbreaks required, yearly, on average, an application of insecticides of over 55 km², in several locations, throughout 81 provinces of Turkey (Figure 6). During the period of 2013–2020, in some provinces, insecticide application has only been conducted in one year, while in some other provinces it has been conducted in two, three or more years. These data have some specific indications for the recent condition for outbreaks in Turkey. Although the grasshopper species are not generally identified and non-locust species are possible, most of the insecticide applications for the period of 2013–2020 are located in the Moroccan locust outbreak areas, i.e., in West Anatolia and Mesopotamian Turkey. This can be considered as an indication of the potential danger of the Moroccan locust for present and the future. Insecticide application in other areas is rare, just in restricted locations and only for one or two years, for the period 2013–2020. Rare application areas are mostly scattered along the one-third of Anatolia, in northern part. From previous publications and personal observation during the last 30 years, we assume that the species outbreaks in these districts are non-locust species mentioned above. In particular, the release of partridges in the provinces of Çorum and Kastamonu in northern Anatolia is consistent with this assumption. Comparison of these recent control activities with earlier records indicates that the potential of outbreak by resident species has increased and may increase further under global warming (see below). Overall, the data indicate that the outbreak potential of the non-locust species requires special attention.

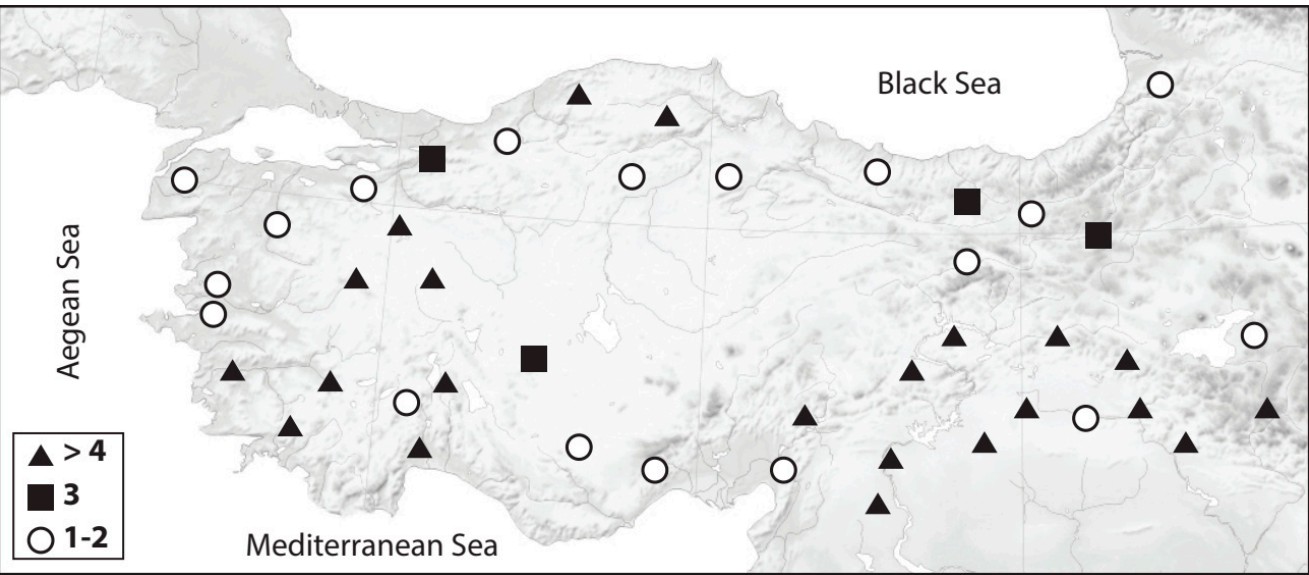

**Figure 6.** Insecticide application by the Directorate of Plant Protection Central Research Institute (DPPCRI) during the period of 2013–2020 throughout Turkey, indicated by the number of years (open circle application in ≤2 years, square in 3 years and triangle in ≥4 years).

## 7. Climate Change and Its Possible Impacts on Locust/Grasshopper Problem in the Near East

Global changes/warming directly or indirectly affect locust and grasshopper species [52–57]. The direct effects are those related to the auto-ecology (e.g., specific thermal thresholds for their development) and therefore to phenology, including changes in life history traits such as the hatching time, instar durations, adult emerging time, reproduction period, egg lying time and number of annual generations [53,54,58,59]. Such phenological changes depend on the resilience and adaptive capacity of the species [54,58–61]. The indirect effects of global warming on insect species are represented by changes in habitat as well as changes in food preferences. Changes in habitat characteristics can be particularly

serious for phytophagous insects, as their presence is correlated with plant communities [53,54]. Locusts, generally phytophagous, may have some preference for a limited plant spectrum or may preferentially feed on some certain plant species. For example, Uvarov [18] reported that *Plantago* spp. and *Hordeum* spp. were determining plants for the presence of the Moroccan locust in western Anatolia. Thus, global warming will produce new conditions for locusts and grasshoppers and adaptation to the new conditions will require high resilience and/or adaptive plasticity, especially for the locally outbreaking species [41,54–57].

Depending on the adaptive capacity of a species, three main consequences may be expected: (i) change of ecological niche/habitat or phenology, (ii) dispersal to new areas with conditions similar to those of the original habitats (range shift) and (iii) extinction, if there is no capacity to adapt to the new conditions and/or to disperse into new areas [52–57]. A change in ecology/phenology of the pest orthopteran species is very likely. If their long-term adaptations provide the capacity for a broad ecological tolerance, they can overcome altered environmental conditions and continue to live in their regular range and outbreak if they meet favorable conditions. In the case of outbreaking species, one issue to be considered in the context of phenological changes is the number of generations per year. The lengthening of the warm period, a consequence of global warming [62,63], can lead to an increase in the number of generations. Shifting their ranges either diffusely over several generations or migrating long distances within a generation can be expected when the previous habitat becomes unavailable [41] or newly available areas appeared [57,61,63]. Regarding outbreaking and invasive locusts, the new conditions produced by global warming may also mean an increase in the available area to be invaded, in addition to their regular ranges [56,57]. Extinction or even range size reduction is less likely for common locusts but may be probable for locally outbreaking non-locust orthopteran. On the other hand, the severity or impacts of global warming/changes are expected to be different depending on geographic location [60,64]. The Mediterranean in general and the Near East in particular are among the places experiencing a high rate of warming [64] and the possible consequences for the species in this area are likely to be more serious.

The possible impact of global changes/warming on outbreaking species can be assessed according to four criteria: (i) phenological changes, including the number of annual generations, (ii) habitat/niche change, (iii) range shift and (iv) extinction. Although each species may be affected differently, a generalization may be possible for the locally outbreaking species, such as those of Barbitistini belonging to *Poecilimon, Isophya* and *Phonochrion*. Unfortunately, studies specifically examining the bio-ecology of the species belonging to these genera are scarce [51] and do not provide a basis with which to evaluate the possible impact of global warming for their future. Some speculations can be made from a phylogeographic point of view regarding the location of their range and habitat preference. These species are univoltine and this feature seems to be fixed evolutionarily; therefore, additional generations are unexpected. However, their hatching, nymph and adult periods may shift to earlier times of year. Outbreaking species of these genera prefer certain vegetation, mainly wet grasslands, and their ranges are mostly restricted to certain fragments in the heterogeneous topography of the Black Sea Basin and of the highlands of East Anatolia [41,65–67]. Furthermore, they are flightless, with a low dispersal capacity, and we believe that invasion of new areas by these species is unlikely. On the other hand, they can vertically shift their range and benefit from the buffer role of altitudinal heterogeneity to overcome the effect of global warming [68]. This is another reason why extinction is also unlikely. These species are likely to experience fluctuation in population density from year to year or at certain years and become harmful when the appropriate combination of ecological factors is met. Thus, they may require special monitoring and necessary control measures must be taken in the event of an increase in population density.

A generalization may also be possible for the locally outbreaking acrid species, i.e., *L. migratoria, H. pterosticha, N. anatolicus, C. italicus, C. barbarus, C. tenuicercis* and *A. labiata*. Different than all others, the last species is found in mountainous habitats and

will not be considered. The other species are mainly associated with anthropogenic or agricultural habitats and prefer a relatively warm climate and moderate altitudes below 1000 m [43]. The first two species are found in humid and irrigated agricultural plains, while the last four in arid plains or arid habitats contiguous to humid plains. These species are known to be univoltine. However, our observations on *H. pterosticha* indicate that it can be multivoltine as nymphs and adults have been observed together at different periods from June to October. From the perspective of the above four criteria, the following statements can be made. The probability of extinction is less likely. Range shifting and an enlargement of the regular range is highly likely as warming provides new opportunities for these species. An additional generation seems less likely for most of the species, but this is probable for *L. migratoria* and *H. pterosticha*. *N. anatolicus*, and three species of *Calliptamus*, namely, *C. italicus*, *C. barbarous* and *C. tenuicercis*, occur in arid and mostly agricultural areas (48). Publications on locust generally report *C. italicus* as the main pest species in the genus; the author observed that these three species reach high population densities in the arid areas adjoining to irrigated plains. Modelling studies [56,57] estimate that *C. italicus* may benefit from global warming and enlarge its range or become a more serious pest in West Palearctic. Estimating the same consequences for *C. barbarous*, *C. tenuicercis*, and *N. anatolicus* will not be an exaggeration as their habitat preferences are quite similar. All these species may become more serious pests as temperatures and agricultural activities increase, and control measures may be an unavoidable necessity.

The two most important species showing real locust characteristics are *S. gregaria* and *D. moroccanus* and they should be considered separately. The first species, the desert locust, live in desertic habitats and the above model cannot be applied. Aridification due to global warming is an expectation for the whole region and especially Mesopotamia [62,64] and this can be considered as new opportunities for the desert locust. The Near East is not an outbreak area for this species, not even in its recession area, but it is on the margins of the invasion area. Larger breeding areas may become available for this locust, offering new opportunities to increase its populations in the case of new invasions. In particularly, enlargement of the dry desert-like areas in Central Anatolia requires special attention in the event of a new plague. Additionally, the inter-mountain plains that develop in Mesopotamia and eastern Anatolia also require caution. However, in any case, preventive or proactive control in the outbreak areas (far from Near East) is more important than the regional measures, which can only be considered during plague episodes.

The four criteria given above can be strictly applied to the Moroccan locust, the most economically important locust species in the region. Uvarov [18], also supported by other studies [40], constitutes a hallmark background for the bio-ecology of the species, to be used as a basis for the four criteria given above. This species is univoltine and a change in this characteristic due to new conditions seems unlikely. However, some climate changes, such as the lengthening of the warm season, can shorten various stages of the life cycle and cause earlier outbreaks. Shift of the distribution area is another parameter of our model. This species occurs in certain altitudes: 400–1000 m altitude in western Anatolia [18] and 200–400 m in Mesopotamia [13]. Thus, dispersing to higher altitudes along the outbreak areas that are suitable for other conditions is likely. However, species composition of vegetation constitutes another determinant of the outbreak. Thus, change in vegetation composition either in outbreak areas or in areas to be invaded further is also important. In addition, the precipitation regime is another significant factor in the ecology of this species, so fluctuation in the precipitation regime will be another determinant for the future outbreaks by the species. However, it should be noted that aridification in the region is an expectation [64] and this may interplay a role in an outbreak. Together, these conditions may further delimit the expansion and outbreaking of this species, as has been the case since the 1960s. As a result, it may become a less important pest, as observed in several other places [40]. Plagues of the species are possible in the case of upsurges during a few successive years. Thus, monitoring the species in local outbreak areas, as has been done

during 2013–2020 by the DPPCRI, especially in the cases of random overlapping of the conditions mentioned above, remains the best practice to prevent possible plagues.

## 8. Conclusions

The above assay includes data gathered from studies in various scientific fields ranging from the humanities to phylogeography and such a rich combination indicates that locust plagues are natural events with a high impact on human life. As witnessed in the 20th century in the Levant, Mesopotamia and Anatolia, every society or state has to pay attention to it in order not to pay the price. In the light of past plagues, it can be stated that the two most important and damaging species were true locusts: *S. gregaria* and *D. moroccanus*. For *S. gregaria*, although the Near East does not include any outbreak area and remains just on the margins of the invasion area, societies living in this region have had to endure tremendous suffering and have paid hundred thousand lives. These two species prevailed in the region for nearly 100 years, between the mid-19th and 20th centuries, but large plagues came to a halt from the 1960s. *S. gregaria* is still a serious pest for North Africa and the Southern Arabian Peninsula, as well as in many other regions of its invasion area, but effective control in its outbreak areas [37,69] keeps the Near East apart of the disaster of this species. *D. moroccanus* was probably the most harmful species especially for West Anatolia and Mesopotamia, particularly in the first half of the 20th century. However, effective control in its outbreak areas/plains scattered in West Anatolia and Mesopotamia subsequently prevented large-scale plagues. As for *S. gregaria*, only a few local swarms were encountered after the 1960s. Although they lack the typical locust characteristics and do not swarm in extensive areas, other species pullulate locally and may cause significant damage in crop fields and pastures, especially in Anatolia. This is the case with *L. migratoria*, *C. italicus*, *C. barbarus*, *C. tenuicercis* and *H. pterosthica*, all from Acrididae, which should be closely monitored for early prevention. Outbreak of species belonging to the Ensifera, namely those of Barbitistini, is an event rarely reported. However, these species may cause significant damage to grasslands and require monitoring. Global changes, especially climate warming, can affect these species in various ways, changing the outbreak probability, as well as the course of invasions and migrations. We can assume that most of these species will benefit from these changes to invade new areas, enlarge their range or increase their population sizes, at the disadvantage of humans. Such global changes may present opportunities for the pest species of Acrididae, both in the irrigated and arid agricultural plains of Mesopotamia and Anatolia. Scientific attention on the part of local authorities is essential not to face such disasters as experienced during the First World War.

**Funding:** This research received no external funding.

**Institutional Review Board Statement:** Not applicable.

**Informed Consent Statement:** Not applicable.

**Data Availability Statement:** Not applicable.

**Acknowledgments:** This review would not have been possible without invitation, encouraging and valuable contributions by Michel Lecoq. Numan Babaroğlu, Mehmet Çulcu and Emre Akci from Directorate of Plant Protection Central Research Institute (Ankara) spent a great time and effort to provide data on grasshopper management for the period 2013–2020. Onur Uluar and Özgül Yahyaoğlu assisted during preparation of the review. Designer Helin Çıplak Palabıyık (helin@wodern.com) prepared the maps. Hasan H. Başıbüyük reviewed the manuscript for linguistic editing. We are grateful to all for their valuable contributions.

**Conflicts of Interest:** The author declares no conflict of interest.

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
