# Peer review of "Locust and Grasshopper Outbreaks in the Near East: Review under Global Warming Context"

_agronomy, doi:10.3390/agronomy11010111_

Round 1

Reviewer 1 Report

Dear Author,

I enjoyed reading this review, but found that some recent papers by our Russian colleagues were not covered. These concern the same locust species including Calliptamus, so they should be included (list below). I think this manuscript would also benefit from a copy edit by a native English speaker; it’s well written, but some use of the passive voice causes some awkward constructions.

Popova, E.N., 2014. The increase in locust threat following climate warming in Russia: calculating estimations. Agro XXI, 7–9.

Popova, E.N., Semenov, S.M., Popov, I.O., 2016. Assessment of possible expansion of the climatic range of Italian locust (Calliptamus italicus L.) in Russia in the 21st century at simulated climate changes. Russ. Meteorol. Hydrol. 41, 213–217.

Latchininsky, A.V., 2017a. Climate change and locusts: what to expect? Proc. Russ. State Hydrometeorol. Univ. 46, 134–143.

Also these papers, though they’re in Russian.

Latchininsky, A.V., Kokanova, E.O., Gapparov, F.A., Childebaev, M.K., Temreshev, I.I., 2015. Locusts and climate change. Bull. Al-Farabi Kazakhstan State Univ. Ecol. Ser. 2/2 (44), 641–648. in Russian with English summary.

Musolin, D.L., Saulich, A.H., 2012. Reactions of insects to recent climate change: from physiology and behavior to distribution area shift. Entomol. Rev. 91, 3–35(in Russian with English summary).

All of the above papers were reviewed by Cullen et al (2017), so that review might also be useful:

Darron A. Cullen, Arianne J. Cease, Alexandre V. Latchininsky, Amir Ayali, Kevin Berry, Jerome Buhl, Rien De Keyser, Bert Foquet, Joleen C. Hadrich, Tom Matheson, Swidbert R. Ott, Mario A. Poot- Pech, Brian E. Robinson, Jonathan M. Smith, Hojun Song, Gregory A. Sword, Jozef Vanden Broeck, Rik Verdonck, Heleen Verlinden and Stephen M. Rogers, From Molecules to Management: Mechanisms and Consequences of Locust Phase Polyphenism. In: Heleen Verlinden, editor, Advances in Insect Physiology, Vol. 53, Oxford: Academic Press, 2017, pp. 167-285.

Well done with the review, and I look forward to seeing an updated version.

*ENDS*

Author Response

Thank you very much processing and reviewing the manuscript. Every comments were considered carefully and the manuscript has been revised accordingly. I should mention that the present paper is a short review paper, and introduction outlines the content, instead of providing a detailed background for every issue in the details. Short background per subheading provided following the subheadings. Thus, I have not add new sentences to the introduction for global warming.

Thank you very much for the comments allowed us to improve the manuscript further.

Sincerely

Battal Ciplak   

Reviewer 2 Report

The manuscript “Locust and grasshopper outbreaks in the Near East: Review under global warming context“, by Battal Çiplak, summarises the history, geography, ecology and economic effects of orthopteran swarm plagues in the Near East.

This review is well designed and researched, written in a clear way, and contains instructive figures. It blends information from historical sources and recent data over the last decades for the region in focus to provide a unique overview which is missing so far in the literature. It also makes use of up-to-date information from the surveillance of different insect swarms. Notably, it covers not only locust species but also some species of Ensifera, the long-horned orthopterans. By the historical and geographic focus, the review will provide a significant contribution to dealing with locust swarming.

As the author states in the introduction, this region deserves specific consideration, and in turn might give insights into pest control from insects, as it differs to the situation in African and Arabian regions. This rationale is convincing, but should be explained in some more detail in the introduction by explaining and referencing what these differences are, and how they affect the occurrence of swarming Orthoptera compared to the other regions (see also below).

Only few minor points should be addressed for clarity:

Specific comments:

l. 11 This statement is rather general - include the important means of control here.

l. 25f Introduction: as the review focusses on global change/warming already in the title, it would be apt to include a short summary of the ecological conditions of the analysed region or the conditions particularly susceptible to expected changes.

l. 31 …in Assyrians and in the New Testament…

l. 37f The introduction here states that “the locust and grasshopper problem in the Near East has other peculiarities, thus require a separate evaluation.“ - Because this introduces the rationale of the present review, the background details are important to be elaborated more and it should be listed what peculiarities are alluded to in the sentence.

l. 47 first bracket is missing

l. 48 long-horned

l. 64 (eg. [6])

l. 68 This is a very interesting point that should be described in some more detail – please include a brief summary what measures this law took against the insects.

l. 70, l. 223f give full names or abbreviations for the species.

l. 77 For the historical summary, it would be interesting to include e. g. original photographs from the swarm invasion endured during a major plague or the dam systems restricting migrations, to give a direct impression of the situation, if available.

l. 82 The history of desert locust plagues…

l. 141 mid of 20th century

l. 226 species identification?

l. 238 N. Babaroğlu? Compare l. 205

l. 454 “provided assisted“? One word seems to be redundant here.

l. 725 Collecting specimen…/ Collection of specimen…

Author Response

(The authors gave the same response as above.)
